

# Micromorphological characteristics of sandy forest soils recently impacted by wildfires in Russia

Ekaterina Maksimova[1,2], Evgeny Abakumov[1,2]

[1]Saint-Petersburg State University, Saint-Petersburg, Russia
[2]Institute of Ecology of Volga Basin, Togliatti, Russia

*Correspondence to*: Ekaterina Maksimova (doublemax@yandex.ru)

**Abstract.** Two fire affected soils have been studied using micromorphological methods. The objective of the paper is to assess and compare fire effects on micropedological organization of soils in a forest steppe zone of the Central Russia (Volga Basin, Togljatty city). Samples were collected in the green zone of Togljatti city. The results showed that both soils were rich in quartz, feldspar. Mica was highly present in soils affected by surface fires, while calcium carbonates were identified in the soils affected by crowns. The type of plasma is humus-clay, but the soil assemblage is plasma-silt with the prevalence of the silt. Angular and subangular grains are the most dominant in soil particulates. No evidence of intensive weathering was detected. There was a decreasing of the porosity in soils affected by fires as consequence of soil pores filled with ash and charcoal.

## 1 Introduction

Fire has an important impact on soil properties as identified by previous works (Certini, 2005; Certini, 2013; Guénon et al., 2013; Mataix-Solera et al., 2011; Jain Theresa et al., 2012; Bergeron et al., 2013; Dymov, Gabov, 2015; Zharikova, 2015; Maximova, Abakumov, 2015). Soil processes in postfire soils environments is quite different from those in soils of natural landscapes or in thechnogenous ones. In general, changes in morphological organization and soil mineralogy are well known in soils after fires produced at high temperatures. After the fire, there is an accumulation of ash on topsoil (Pereira et al., 2014), leaching of some nutrients into deeper horizons (Bodi et al., 2014), over-compaction of the surface and accumulation of crusts, transformation of soil structure (Mataix et al., 2011).

Micromorphological methods are known as useful tool for investigation of soil transformation under natural and human-impacted conditions (Stoops, 2009). Methodology of classical micropedology provides with required information about soil development in micro level, such as changes in fine earth composition and soil plasma evolution (Kubiena, 1938, 1967; Stoops and Eswaran, 1986). These methods are widely used for analysis of soil paleo processes (Sedov et al., 2013), soil restoration on post-mining environments (Abakumov et al., 2005), soil elementary process in different environments (Lebedeva et al., 2010; Abakumov et al., 2013) and specification soil classification aspects (Kubiena, 1967). Micromorphological investigations related to fire effect of soil crusts and fine earth (Greene et al., 1990) and aggregate dynamics in post-fire soil (Mataix-Solera et al., 2002) have been undertaken. However, the micromorphological methods have never been applied to study post-fire soils transformation in the Russian wild-fire environments. Moreover, this paper deals with comparison of different post-fire scenarios (surface and crown fires) and provide 5-year monitoring of fire affected soils and ecosystems in a whole. Few researches were carried out about this topic.





The objective of this work was to characterize the micromorphological indices of microstructure transformation
in soils affected by different types of wild-fire, as compared microstructure of mature, unaffected soil of pine
forests in Central Russia.
**2 Materials and methods**
**2.1 Study area**
The study was conducted in Samara region situated near the Volga River, in the central part of European Russia,
Samara region (N 58°39'44.55"; E 39°17'48.95", 179 m asl). The extremely hot weather in summer of 2010 in
Russia (especially the most difficult situation was on whole Russian European area and also Ukraine and Eastern
Europe) resulted in drought and eventually catastrophic forest fires on the vast territories of European and Siberian
Russia. The forest fire studied occurred in 2010 and affected more than 8000 hectares. Fire severity was very high.
Parent material is composed by old (Pleistocene) alluvial-dune landscape.
The affected ecosystem is characterized by a forest-steppe environment with higher pedodiversity. The vegetation
was composed mostly by pine forests *Pinus sylvestris L.* There are xerophyte species at dry locations like *Veronica*
*spicata L., Sedum acre L., Antennaria dioica L., Calamagrostis epigeios (L.) Roth, Centaurea marschalliana*
*Spreng* and hardwoods (*Quercus robur L., Betula pendula Roth., Populus tremula L.*) in more humid conditions.
The herbaceous vegetation consists of rhizomatous and loose-bunch gramen *(Bromus inermis Leyss., Elytrigia*
*repens L.,* some species of *Poa*, and *Agrostis canina L.)* at post-fire plots.
A variety of Luvisols and Chernozems prevails in the watershed sections (Nosin, 1949; Vasil'eva, Baranova, 2007;
Abakumov, Gagarina, 2008; Abakumov et al., 2009; Urusevskaja et al., 2000). Whereas, Calcaric Chernozems
(southern type) dominate in the south of the region (steppe zone of the Samara region), accompanied with some
polypedons of Kastanozems (Nosin, 1949). Soils of investigated area are sandy and sandy loam textured. In the
studied area, soils were classified as sandy loam soils on Late Pleistocene alluvial Volga sands – Protoargic
Arenosols according to World reference base (2015), and they have weak features of illuvial phenomena without
formation of separate horizons. Sand content in these soils is 70.5-86.4%; clay content is 0.3-2.6 %.
Three soil pits were sampled in two different fire affected areas: one in a site affected by a surface forest fire and
another affected by a crown fire. A sample was collected in an unburned area to serve as control. Soil types and
vegetation were the same. Soils were sampled as fast as possible after the removal of a state of emergency from
the territory in summer of 2010, and also during the period 2011-2015. Three soil pits were sampled at each studied
area made at each plot. Undisturbed soil samples were collected using Kubiena-type boxes of 5x3.5x1.5 cm sizes
at the depth of 0-10 cm and taken to the laboratory.
**2.2 Micromorphological analysis**
Soil samples were air-dried and being passed through a 1-mm sieve. Fine sections of soil material were prepared
form micro monoliths of soils, sampled in field. Samples were dried and saturated with resin.
Thin sections were investigated with use of polarization microscope Leica DFC 320 in transmitted light and
crossed nicols. The following soil micromorphological indexes have been investigated: soil microfabric, spatial
arrangements of fabric units, soil particles distribution, elements of microstructure and character of organic matter.



Terminology, used in this paper are published by Stoops (2003), also by Gagarina (2004) manuals and Gerasimova
(2011) review, where details of micro organization of soil were described in details.
**3 Results and discussion**
**3.1 Soil profile analysis and physico-chemical properties**
The soil profile organization can be described as: A – AC – C. The A horizon has a black layer due the deposition
of an ash and charcoal layer on soil surface, contrary to the observed in the control plot (fig. 1,2).

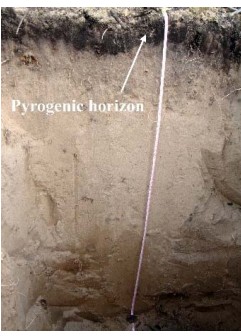

**Figure 1: Fire affected soil.**        **Figure 2: Control soil.**
Data on the general characteristics of the soils are given in Table 1.
Organic matter is lost from the surface horizons of the soil, which is related to the destruction of the organic
horizons, the mineralization of root residues, and the almost complete absence of fresh plant waste, which could
be a material for humification. In 2010, the content of organic matter in the ash on the soil surface after the surface
fire (2.31±0.27%) was lower than that after the crown fire (3.19±0.19%). A similar tendency was observed in the
pyrogenically transformed humus-enriched horizons: the content of organic carbon was 1.21±0.50% after the
surface fire and 1.42±0.31% after the crown fire. Thus, a surface fire, which leads to the complete burning out of
the litter and the upper horizons, results in larger losses of organic matter.
The acidity in the upper horizons of the burnt soils decreases significantly, and the burnt litters have an alkaline
reaction (pH 7.9-8.0), while the lower horizons have a weakly acid reaction close to that in the corresponding
horizon of the undisturbed forest soil (pH 5.7-5.9). The increase in the pH values of the soils after fires is related
to the fact that the water-soluble ash components penetrate into the soil and saturate the soil exchange complex
with alkaline-earth elements, which shifts the reaction toward the neutral value. No differences in the pH changes
between the effects of the crown and surface fires were observed in the first year of the study. The reaction of the
parent rock is similar in all three plots and is characterized as weakly acid. The partial or sometimes complete
mineralization of organic residues because of pyrolysis resulted in the synchronous input of ash elements onto the
soil surface and into the litter, which neutralized the organic acids arriving in the soil solution during the
decomposition of the litters. It is therefore obvious that the higher the ash yield (i.e., the more intensive the fire),
the more complete and active the neutralization of the litters.
A year after the fire, the pH of the burnt litters decreased from 7.9-8.0 (in 2010) to 6.4-6.6 (in 2011), and its
absolute values approached the control level. This is easily explicable: rain and snowmelt waters almost completely



removed the soluble ash components for a year; i.e., the alkali elements were removed from the ash at the fire
sites.
One-Way Anova test has shown that significant differences were revealed for carbon content between surface fire
and crown fire ($p < 0.02$) and also between control and surface fire plot ($p < 0.01$). As for silt-and-clay fraction there
were differences only between control and surface fire ($p < 0.04$). The same situation was characteristic for sand
fraction ($p < 0.02$).
The WRB system (World reference base…, 2015) does not have any horizons or diagnostic parameters of
pyrogenic soils. But WRB system has Pretic horizon as dark, high content of organic matter and phosphorous, low
biological activity, high contents of exchangeable calcium and magnesium, with remnants of charcoal and/or
artefacts. A pyrogenic horizon with abundance of charcoal is formed after wildfires. It can resist to degradation
when vegetation has not started to recover yet at burned places and charcoal has not started to redistribute while
erosion and infiltration processes. However, black carbon decomposition is controversial, and there are different
views about this issue. Some studies argue that black carbon decomposes very slowly (Liu et al., 2008) or
practically non-degradable (González-Pérez et al., 2004), while others show that it successfully affected by
chemical (Cheng C.H. et al., 2006, 2008) and microbial (Knicker et al., 2013; Marschner et al., 2008) oxidation.
The assumption of black carbon complete stability in soils is doubtful because its content varies considerably in
different soils that is explained not only by a difference of pyrogenic activity in different natural zones, but also
by a difference of humidity (Nguyen, Lehmann, 2009) and temperature (Cheng C.H. et al., 2008; Nguyen et al.,
2010), by various physical and chemical soil characteristics, different biological activity and land use practices
(Czimczik, Masiello, 2007).
**3.2 Micromorphological characteristics**
Differences of postpyrogenic and unburned soils are well-shown in morphological, chemical, physical and
biological properties of horizon A. Morphological organization of solum in burned soils differ from unburned in
a number of parameters: wide distribution of charcoal pieces, absence of litter and its transformation into ash that
is a mix of mineral soil components, burned-down plant residues, and small pieces of charcoal (Bodi et al., 2014;
Pereira et al., 2014, 2015); an also reduction of the humic horizon's depth. There is a soil erosion because of
rainfall characterized by decrease of black surface horizon thickness after several years of investigation as observed
in early studies (Francos et al., 2016).
Data on soil micro morphological features are presented in table 2, fig. 3. The results showed that elementary
assemblage of crown fire soil is plasma-silt (with small content of plasma and prevalence of silt particles). In case
of the soil affected by surface fire, the type of assemblage is plasma-silt, which is a result of the accumulation of
humus-type organic matter. In both cases the type of the plasma classified as humus-clay. The structure of all 3
soils investigated is crumb, inherited from the previous stages of soil formation. There are not evident features of
mass aggregation in the fire affected soils. The type of the microstructure of all the soils is angular blocky or
subangular blocky, which is caused by low intensity of current weathering in soil mass. The particle shape is
subidiomorpic or idiomorphic in cases investigated with weak or medium degree of corrodness.
Mineral composition of the soils can be described as follows: uncoloured minerals, quarts, orthoclase and
carbonates (crown fire); quarts and orthoclase with many crystals of muscovite with absence of carbonates (surface
fire) and predominance of uncoloured minerals with quarts and muscovite in case of the control plot. The



possibility of new minerals formation under the strong heating effect has been reported in previous works (Nobles,
2010; Leon et al., 2014), however, carbonate accumulation is not resulted from heating process in scenario of the
crown fire. In some arrangements, soil microstructure can be classified as skeletal, which is caused by high content
of not weathered soil particles. No evidences of current weathering (alteration) have been fixed, but some pores
infilling were recognized as a result of ash and charred material accumulation in fire affected soils (fig. 4). Not
decomposed organic tissues and residues infills the porous media in these soils. This is result of increment of raw
forms of organic material in burned soils. Organic matter under the effect of fire was polymorphic (fig. 4, a, b)
(Stoops, 1986), while it is monomorphic in the soils of the control plot. The porous media infilling after the fires
was described previously by Nobles (2010), however in this case was an accumulation of Mn-Fe enriched
materials. Balfour and Woods (2007) observed similar results in fire affected soils. So, taking our data into account
it is possible to conclude that infilling of porous media by material of different composition is typical in burned
soils. Decreasing of the porous media area was also described as result of accumulation of ash (Balfour, Woods,
2007), and this explanation of porous media infilling is more appropriate to our case. Investigated soils
characterizes by developed system of porous media, this is important in sense of heating penetration into soil and
soil sustainability to heating. Soils investigated are not so sustainable to heating than clay textures ones, where the
porous media is not so developed. Porous, cracks and other forms of space give the possibility to the combustion
products penetrate into the soil and affect the PAHs and other products accumulations.

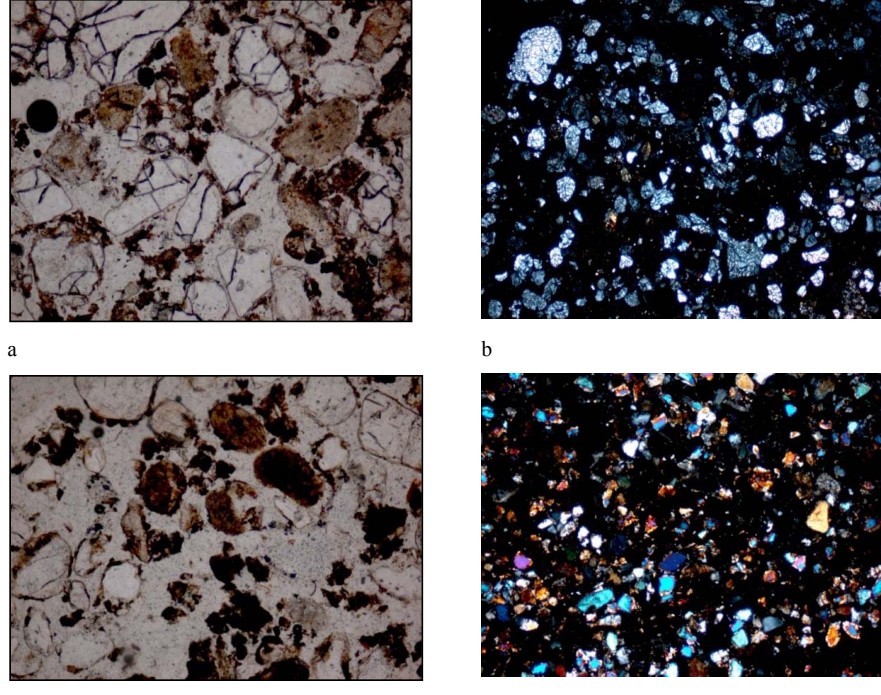

a                                          b

c                                          d



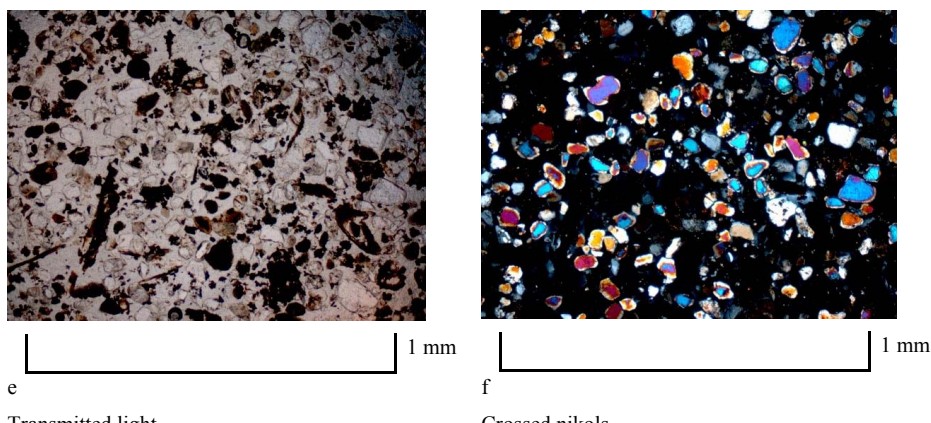

|  | |
|---|---|
| 1 mm | 1 mm |
| e | f |
| Transmitted light | Crossed nikols |

**Figure 3: Thin sections. a, b – crown fire, c, d- surface fire, e, f- nature plot. Left column if transmitted light, right**
**column – crossed nikols.**

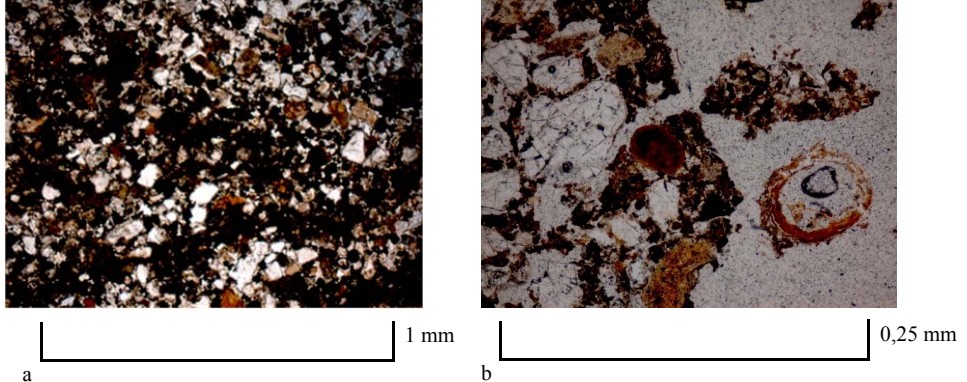

|  | |
|---|---|
| 1 mm | 0,25 mm |
| a | b |

**Figure 4: a – humons (organic particles with not humified organic matter), transmitted light; b – organic tissues of not**
**decomposed remnants.**
The type of fire affected the soil organic micromorphology. The quantity of not humified organic matter in the
burned soils, especially in the samples affected by surface fire was identified as increased in burned soils (fig. 4,
a). It is also evident that the size of organic residues is higher in fire affected soils than in mature ones. In mature
soils transformation and humification of organic matter is gradual process, while new, relatively fresh not
decomposed organic matter has come into upper soil horizons in fire affected areas (Gagarina, 2004).
**4 Conclusions**
The 2010 catastrophic natural fires in the urban forests of Tol'yatti resulted in the formation of pyrogenically
transformed soils, the morphological parameters and the main chemical and physical properties of which
significantly differ from those of the undisturbed soils.
The burnt soils differ from the control soil on the macromorphological level only in the upper part of the profile,
where the litter is transformed to ash identified as a dim-gray organomineral mixture. Processes of soil erosion are
clearly manifested a year after the fire under the effect of precipitation and the illuviation of organic matter to the
medium part of the profile and will probably continue for several years.



The fires significantly affect the physicochemical and chemical properties of the soils. However, the effect of fires
on the properties of the studied soils usually does not spread deeper than 10 cm.
The results of this work showed that mineral composition of all the soils studied is presented by quartz, feldspar
(orthoclase); in case of surface forest fire there was more mica (muscovite), and calcium carbonates appeared in
soils affected by crown forest fire, the reasons of this compound accumulation are still not well understood. The
type of plasma is humus-clay, but the soil assemblage is plasma-silt with the prevalence of the silt. Angular and
subangular grains form the main soil carcass and no evidence of intensive weathering alteration has been revealed.
At the same time, decreasing of the porous media was recognized as a main soil development process after the
fires. This is result of soil porous infilling by ash and charred organic material of different nature: some organic
remnants (tissues) come into the porous media after fire and some transformed, coaled and dark coloured part also
appears in post fire horizons. Partially decomposed fire affected particles of soil organic matter accumulate in
postfire soils, especially in soil porous media, which is a result of the soil organic matter accumulation and
transformation in postfire environments.
**Acknowledgements**
The authors thank the director of Institute of Ecology of the Volga River Basin of the Russian Academy of Sciences
(IEVB RAS), Dr. professor G.S. Rosenberg, to the deputy director for science of IEVB RAS, Dr. professor S.V.
Saksonov and to Dr. S.A. Senator for the help in the organization of work and research support and MSc Rita
Lazareva for her kind assistance with thin sections description.
This work was supported by Russian foundation for Basic research, projects 14-04-32132, 15-34-20844 and Saint-
Petersburg University grant № 1.37.151.2014.

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



1    **Table 1. Morphological features and general properties of soils, ± after the mean value means SD**

| Horizon | Depth, cm | Colour | Soil humidity, % | pH | Total organic carbon, % | Clay content, g·kg⁻¹ | Sand content, g·kg⁻¹ |
|---|---|---|---|---|---|---|---|
| Surface fire | | | | | | | |
| Burned litter | 0-5 | 10 YR 3/2 | 2.85±0.79 | 8.0±0.06 | 2.31±0.27 | 20.0 | 705.0 |
| AY | 5-14 | 10 YR 4/2 | 1.45±0.19 | 6.2±0.32 | 1.21±0.50 | 25.0 | 807.0 |
| AC | 14-27 | 7.5 YR 5/4 | 1.38±0.36 | 6.0±0.21 | 0.75±0.48 | 26.0 | 835.0 |
| AC | 27-36 | 10 YR 6/4 | 1.02±0.27 | 5.8±0.21 | 0.31±0.13 | 19.0 | 852.0 |
| AC | 36-53 | 10 YR 6/4 | 0.98±0.42 | 5.3±0.31 | 0.22±0.05 | 24.0 | 866.0 |
| C | 53-73 | 7.5 YR 5/4 | 0.69±0.05 | 5.7±0.21 | 0.24±0.10 | 13.0 | 864.0 |
| Crown fire | | | | | | | |
| Burned litter | 0-3 | 10 YR 3/2 | 2.37±0.36 | 7.9±0.12 | 3.19±0.19 | 17.0 | 720.0 |
| AY | 3-10 | 10 YR 4/2 | 1.43±0.35 | 5.9±0.38 | 1.42±0.31 | 20.0 | 788.0 |
| AC | 10-15 | 7.5 YR 5/2 | 0.86±0.20 | 5.9±0.25 | 0.78±0.07 | 17.0 | 852.0 |
| AC | 15-24 | 10 YR 7/4 | 1.11±0.63 | 5.9±0.36 | 0.26±0.07 | 6.0 | 862.0 |
| C | 24-44 | 10 YR 6/3 | 0.52±0.05 | 5.7±0.12 | 0.14±0.05 | 4.0 | 867.0 |
| C | 44-64 | 10 YR 6/3 | 0.49±0.03 | 5.9±0.25 | 0.12±0.05 | 9.0 | 868.0 |
| Control | | | | | | | |
| Litter | 0-7 | - | 5.92±2.27 | 6.5±0.10 | nd | nd | nd |
| AY | 7-10 | 10 YR 4/2 | 1.60±0.47 | 6.3±0.06 | 1.94±1.35 | 21.0 | 787.0 |
| AY | 10-14 | 10 YR 6/4 | 0.78±0.15 | 6.2±0.23 | 0.78±0.33 | 18.0 | 837.0 |
| AC | 14-23 | 7.5 YR 3/2 | 0.78±0.42 | 6.1±0.23 | 0.33±0.10 | 13.0 | 867.0 |
| AC | 23-33 | 2.5 YR 8/6 | 0.42±0.01 | 5.9±0.20 | 0.15±0.02 | 4.0 | 888.0 |
| C | 33-50 | 2.5 YR 8/6 | 0.33±0.02 | 5.7±0.12 | 0.21±0.03 | 7.0 | 891.0 |
| C | 50-70 | 2.5 YR 8/6 | 0.32±0.03 | 5.8±0.31 | 0.15±0.05 | 3.0 | 895.0 |

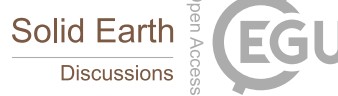

1    **Table 2. Micromorphological characteristics of the soils under investigation**

| Horizon | Sampling depth (cm) | % | Character of distribution | Size, mm max | Size, mm min | Size, mm dominant | Size sorting | Particle shape | Rounding | Corrodness | Mineral composition | Type of plasma | Elementary assemblage | Plant residues % | Plant residues Decomposition degree | Structural separates | Voids | Soil microstructure | Coprolites Size, mm | Coprolites Description |
|---|---|---|---|---|---|---|---|---|---|---|---|---|---|---|---|---|---|---|---|---|
| Crown forest fire | | | | | | | | | | | | | | | | | | | | |
| Apir | 0-15 | 40-50 | Uniform | 0.07 | 0.01 | 0.04 | good-medium | subidiomorphic-xeromorphic, regenerate (some minerals in one) | rounded | weak | colorless minerals are prevail: quartz, feldspar (orthoclase), carbonates (calcispar) | humus-clay | plasma-silt (little plasma, most of all – silt particles) | 10 | weak-medium | aggregates | pores | pore structure with separated aggregates | | |
| Surface forest fire | | | | | | | | | | | | | | | | | | | | |
| Apir | 0-14 | 40-50 | Uniform | 0.07 | 0.01 | 0.01 | medium-weak | idiomorphic, subidiomorphic, xeromorphic | rounded-not rounded | medium | a lot of specular stone (muscovite), quartz, feldspar (orthoclase) | humus-clay | plasma-silt | 5 | weak | aggregates | pores | pore structure with separated aggregates | 0,4 | brown, roundish, harsh with minerals (calcispar) |
| Control | | | | | | | | | | | | | | | | | | | | |
| O | 0-20 | 40-50 | Uniform | 0.09 | 0.01 | 0.02 | good-medium | idiomorphic, subidiomorphic, xeromorphic | rounded-not rounded | weak | colorless minerals are prevail: quartz, feldspar (orthoclase), specular stone (muscovite) | humus-clay | plasma-silt (little plasma, most of all – silt particles) | 30 | weak-medium-strong strong is prevail | aggregates | pores | pore structure with separated aggregates | | |

