# Peer review of "Micromorphological characteristics of sandy forest soils recently impacted by wildfires in Russia"

_Solid Earth, 2016_

## Short Comment (SC1) · 2 Feb 2017

This article contains relevant material on the pyrogenic soils changes in the forest-steppe zone of the European part of Russia. We have some comments to this article. 1. It is not clear how much time has passed since a fire to sampling for micromorphology? 2. Horizons designations differ in the text and tables. 3. It is not clear from where the Âńburned litterÂż contains 72% sand??? It's not litter, it pyrogenic horizon (Q or Apyr). 4. The data on the three sites, but pictures only for two plots??? It requires explanation. 5. Designations horizons in Table 1 do not correspond to the descriptions in the text. 6. Suspicious little carbon pyrogenic horizons? This is not typical for pyrogenic horizons 7. In the table, too, the carbon content is better to present at g kg-1 8. A lot of technical

errors: Togliatti in three places spelled differently (in reference to the Institute, in the abstract and text). Page. 4, line 15, left initials in Cheng et al., 2006. Number 3 of reference list presented on Russian. 9. It is necessary to strengthen the discussion. This is the weakest section.

———————————————

---

## Short Comment (SC2) · 8 Feb 2017

The work reveals new significant data on micromorphological composition of postfire soils. Soil cover of Tol'yatti city has been significantly altered due to catastrophic fires in 2010 and resulted in formation of pyrogenically transformed soils. The results obtained expand the scope of understanding of pyrogenic soils and their physical and chemical properties. In my opinion more analytical data on soil physical and chemical properties have to be added for the text body of the manuscript to achieve more evident connections with micromorphological characteristics.

---

## Referee Comment (RC1) · A. Tsibart (Referee) · 5 Mar 2017

The research paper presents novel and interesting results and corresponds to the scope of the Solid Earth. Paper is well structured and conclusions are well supported by the results.

Some details could be added:

1. In the results and discussion section results on the soil carbon, pH and silt and clay analysis are provided. Which methods were used for these analyses? Were all the samples from years 2010-2015 analysed for these parameters? How many samples were used in each analysis? Why is it important to include these soil properties in the

paper devoted to micro morphological properties?

2. Also some details could be added about ANOVA test. How it was conducted? Which software was used? Was the statistical sample checked for the normality? How many samples were included in ANOVA?

3. Fig.1 provides the pictures of control and fire-affected picture. Which fire was it – crown or surface fire? Maybe one more picture could be provided in order to show all 3 investigated plots.

---

## Referee Comment (RC2) · Anonymous Referee #2 · 7 Mar 2017

This is a very good paper about the consequences of wildfires in the soils in Russia. The paper is centered in the changes at a micromorphological characteristics. This is an important topic that can have consequences at a hydrological level. The authors explain how the pores can be filled by ashes or fine carbon particles.

The authors said that the consequences are only in the first centimeters and aslo they detected erosion processes but this erosion is not quantify.

Could the authors find changes in 2010 and 2011?. It is not clear if this first horizon that was analyzed in 2010 and were the changes were more evindent still are present in 2011 or it has dissapear by erosion. Maybe the erosion processes took place due to the fill of the pores of the first centimeters.

What the authors mean when they said: Processes of soil erosion are clearly manifested...

---

## Author Comment (AC1) · 14 Mar 2017

Dear Reviewer! Thank You very much for your comments to my manuscript. I have changed the manuscript according to your suggestions, these changes are highlighted by yellow color, the following comments I have made to Your recommendations:

1) the information about year of sampling for micromorphology was added to chapter 2.1 Study area – Undisturbed and postpyrogenic soil samples were collected in 2011 using Kubiena-type boxes... 2) horizons designations were corrected and there are no differences between horizons designations in text and tables. 3) pyrogenic horizon Apyr after crown forest fire contains 72% of sand. 4) the picture of the third plot has been added to the text. 5) a depth of pyrogenic horizons was established in accordance

with morphological description in a field. The depth of litter horizon in control soil is not so big that is why the depth of pyrogenic horizon after relatively fast wildfires is little. 6) a unit of carbon content was corrected in the table 1 and stated in g kg-1. 7) technical errors such as spelling of Togljatty, corrections in reference list were corrected. 8) several changes were made in results and discussion section.

Thank You very much!

With kind regards, Corresponding author Ekaterina Maksimova

14-03-2017

Please also note the supplement to this comment:
http://www.solid-earth-discuss.net/se-2016-173/se-2016-173-AC1-supplement.pdf

**Supplement:**

[revised manuscript text omitted]

**3 Results and discussion**

**3.1 Soil profile analysis and physico-chemical properties**

The soil profile organization can be described as: Apyr (or O in case of control) – AY – AC – C. The Apyr horizon is a black layer due the deposition of an ash and charcoal layer on soil surface, contrary to the observed in the control plot (fig. 1,2,3). A wide distribution of coal pieces, a total absence of forest floor remnants and its transformation into ash was diagnosed in 2010 immediately after fires. At the beginning of the research (2010), thick black horizons were observed on the soil surface, while in the summer of 2011, they were present as only a thin layer on the surface. This testifies to the influence on erosion, as the soil surface has been affected by precipitations after the disappearance of forest floor (Robichaud, 2005; Vieira et al., 2014; Delwiche, 2009).

[Figure]

[Figure]

[Figure]

**Figure 1: Crown fire affected soil.** **Figure 2: Surface fire affected soil.** **Figure 3: Control soil.**

Data on the general characteristics of the soils are given in Table 1.

Organic matter is lost from the surface horizons of the soil, which is related to the destruction of the organic horizons, the mineralization of root residues, and the almost complete absence of fresh plant waste, which could be a material for humification. Humus degradation of the upper horizons was clearly visible by means of the ignition loss value. Ignition loss was more than 20.00% in the upper layer in the control plot (table 1), but only

5.45% in the crown fire and 5.68% in the surface fire. In 2010, the content of organic matter in the ash on the soil surface after the surface fire (2.31±0.27%) was lower than that after the crown fire (3.19±0.19%). A similar tendency was observed in the pyrogenically transformed humus-enriched horizons: the content of organic carbon was 1.21±0.50% after the surface fire and 1.42±0.31% after the crown fire. Thus, a surface fire, which leads to the complete burning out of the litter and the upper horizons, results in larger losses of organic matter.

The acidity in the upper horizons of the burnt soils decreases significantly, and the burnt litters have an alkaline reaction (pH 7.9-8.0), while the lower horizons have a weakly acid reaction close to that in the corresponding horizon of the undisturbed forest soil (pH 5.7-5.9). The increase in the pH values of the soils after fires is related to the fact that the water-soluble ash components penetrate into the soil and saturate the soil exchange complex with alkaline-earth elements, which shifts the reaction toward the neutral value. No differences in the pH changes between the effects of the crown and surface fires were observed in the first year of the study. The reaction of the parent rock is similar in all three plots and is characterized as weakly acid. The partial or sometimes complete mineralization of organic residues because of pyrolysis resulted in the synchronous input of ash elements onto the soil surface and into the litter, which neutralized the organic acids arriving in the soil solution during the decomposition of the litters. It is therefore obvious that the higher the ash yield (i.e. the more intensive the fire), the more complete and active the neutralization of the litters.

The group humus composition (Cha to Cfa ratio) of the upper soil layers changed as a result of the fires (Table 1). Some authors have noted an increase in humic acid content and a decrease in the carbon-to-nitrogen ratio (Abakumov and Frouz 2009; Efremova and Efremov 2006). On the contrary, the appearance of the most aggressive fractions presented by fulvic acids was recorded in other studies (Dobrovol'skij 2002). In our case, the litter of the control plot was characterized by a fulvic-humic type of humus (for the other horizons, humic-fulvic) and an increase in humic acids, which was especially strong after the surface fire, as characteristic for postpyrogenic soils. A reduction of the Cha/Cfa ratio due to new plant litter was observed in the following years. An increase in humic acid content was also observed in the humus horizon.

[revised manuscript text omitted]

---

## Author Comment (AC3) · 14 Mar 2017

Dear Referee! Thank You very much for your comments to my manuscript. I have changed the manuscript according to your suggestions, these changes are highlighted by green color, the following comments I have made to Your recommendations:

1) soil carbon, pH and silt and clay were investigated using recommendations of monography "Methods of Soil Analysis. Part 3 Chemical Methods" (1996). An appropriate txt was added to the article in section 2.1 Study area. 2) all samples of the period 2010-2015 were analysed but in this paper only data of 2010 were represented as the most informative. 3) the measurements of studied parameters were performed in triplicate – this information was added to the text. 4) data of main soil properties that

have been showed are necessary in order to characterize studied soils in whole and to describe soil organic matter that was diagnosed by means of micromorphological methods. 5) the normal distribution of the data was verified previously, and analysis of variance (ANOVA) and a post hoc test were conducted using SIGMAPLOT 8.0 software with the aim of comparing differences between plots (site effect). Differences were considered significant at $p < 0.05$. Three values of each sample were included in ANOVA. 6) following your kind advise, one more picture of studied plot was added to the text and a type of fire was showed.

Thank You very much!

With kind regards, Corresponding author Ekaterina Maksimova

14-03-2017

Please also note the supplement to this comment:
http://www.solid-earth-discuss.net/se-2016-173/se-2016-173-AC3-supplement.pdf